# Structure, Properties, and Phase Transformations of Water Nanoconfined between Brucite-like Layers: The Role of Wall Surface Polarity

**DOI:** 10.3390/ma15093043

**Published:** 2022-04-22

**Authors:** Alexey A. Tsukanov, Evgeny V. Shilko, Mikhail Popov

**Affiliations:** 1Center for Computational and Data-Intensive Science and Engineering (CDISE), Skolkovo Institute of Science and Technology (Skoltech), 121205 Moscow, Russia; 2Institute of Strength Physics and Materials Science of SB RAS, 634055 Tomsk, Russia; 3Institut für Mechanik, Technische Universität Berlin, 10623 Berlin, Germany; mpopov@fastmail.fm

**Keywords:** nanoconfined water, nanopore, interface, surface polarity, phase transformation, layered metal hydroxide, high-pressure crystallization, molecular dynamics

## Abstract

The interaction of water with confining surfaces is primarily governed by the wetting properties of the wall material—in particular, whether it is hydrophobic or hydrophilic. The hydrophobicity or hydrophilicity itself is determined primarily by the atomic structure and polarity of the surface groups. In the present work, we used molecular dynamics to study the structure and properties of nanoscale water layers confined between layered metal hydroxide surfaces with a brucite-like structure. The influence of the surface polarity of the confining material on the properties of nanoconfined water was studied in the pressure range of 0.1–10 GPa. This pressure range is relevant for many geodynamic phenomena, hydrocarbon recovery, contact spots of tribological systems, and heterogeneous materials under extreme mechanical loading. Two phase transitions were identified in water confined within 2 nm wide slit-shaped nanopores: (1) at *p*_1_ = 3.3–3.4 GPa, the liquid transforms to a solid phase with a hexagonal close-packed (HCP) crystal structure, and (2) at *p*_2_ = 6.7–7.1 GPa, a further transformation to face-centered cubic (FCC) crystals occurs. It was found that the behavior of the confined water radically changes when the partial charges (and, therefore, the surface polarity) are reduced. In this case, water transforms directly from the liquid phase to an FCC-like phase at 3.2–3.3 GPa. Numerical simulations enabled determination of the amount of hydrogen bonding and diffusivity of nanoconfined water, as well as the relationship between pressure and volumetric strain.

## 1. Introduction

Solid and liquid phases confined to a volume of several atomic sizes (nanoconfined matter) may significantly or even qualitatively differ from the corresponding bulk phases. This is valid both for nanosized particles or droplets and for liquids or solids confined within nanosized pores or gaps [1,2,3,4,5]. Under conditions of nanoconfinement, many kinetic and thermodynamic processes—including chemical reactions—demonstrate extraordinary behavior not observed in bulk conditions [6,7,8]. This causes both nanoconfined matter and processes occurring under low-dimensional constraints to be of increasing interest in a variety of scientific and industrial areas [1,9,10]. For example, nanoconfined hydrides are among the most promising solutions for hydrogen storage [11]. Confined ionic liquids are used in promising supercapacitors (electrochemical devices that store energy through the reversible adsorption of ionic species) with enhanced capacitance and the capability of fast charging and discharging [12,13]. Nanoconfinement-based methods are used for the analysis of long biomolecules [14,15], as well as for studies of hydrocarbon-recovery-related problems using nanofluidic devices [16,17]. Matter in nanoconfined conditions is found everywhere in geological media, thereby making nanoconfinement effects essential for the proper understanding of many phenomena in geoscience, including the physical and mechanical properties of saturated porous media [18], formation of superhydrated phases of minerals [19,20], dehydration processes of subducting slabs [20], deformation and rupture of heterogeneous geomaterials [21], and lubrication effects under high-pressure conditions [22,23]. Confinement [22] and nanoconfinement [23,24,25,26,27] strongly influence the mechanical and frictional properties of lubricants in highly loaded contacts.

Taking into account nanoconfinement-related effects is important in solving many materials science and nanotechnology problems, including the development of functional nanomaterials (e.g., molecular sieves and ion-selective membranes [28,29,30,31,32,33]), bionanosensors [34], highly ordered mesostructured composites with unusual mechanical and structural properties [35], auxetic materials [36], efficient sorbents [37], etc. [38,39,40]. Additionally, in a variety of cases, nanoconfinement provides favorable conditions for the catalysis of chemical reactions [41,42,43].

It is well known that confinement of thin layers has a strong effect on the phase state and phase transformations of a substance. Water confined in slit-shaped nanopores of 7 Å width between two graphite plates shows solid-like behavior, whereas in pores of 9 Å width or more this effect is not observed [44]. A similar effect (formation of 2D ice crystals) was recently studied for water bilayers between mica and graphene [45,46]. Moreover, the mechanism of nanoconfinement-induced freezing of liquids was experimentally investigated for cyclohexane, octamethylcyclotetrasiloxane, and toluene [47]. It was found that when the nanopore width is below a certain critical value—about six molecular layers of confined liquid—a liquid-like–solid-like phase transition occurs. Another example is argon; when confined between atomically flat graphene nanosheets, argon does exist in a solid phase, with mixed hexagonal close-packed (HCP) and face-centered cubic (FCC) structures, at pressures significantly lower than the crystallization pressure of argon in a bulk system [48].

Water is among the most interesting subjects for the study of nanoconfined matter, due to its ubiquitous presence and importance for our planet. Even in a bulk state, water shows an unusually large variety of structural forms (including many phases of ice [49]), and has some well-known anomalous features. Confined water plays a key role in many practically significant processes, such as the functioning of biological molecular machines [50], selective ion filtration through biological channels [51], functioning of porins, and changes in the properties of biological components in soft nanoconfinement [52].

The character of the interaction between water and the confining medium is largely determined by the hydrophobicity/hydrophilicity of the pore walls which, in turn, depends on both the atomic structure of the surface and the distribution of electric charge between surface groups of atoms, i.e., surface polarity [53,54,55].

The present study is devoted to the question of how surface polarity can influence the network structure and the density of hydrogen bonds in nanoconfined water, as well as its mechanical properties at extremely high pressures. To carry out the analysis, we used molecular dynamics (MD) simulation with atom-level models. We considered brucite-type layered metal hydroxides as the nanopore wall material. This is a widespread natural mineral class defined by the chemical composition Me^(II)^(OH)_2_, where Me^(II)^ is a divalent metal [56] or a mixture of metals [57]. The MD model allowed us to study the properties and structure of water trapped in the interlayers of typical layered hydroxide minerals under different pressures, and to directly compare the behavior of nanoconfined water under hydrophilic and hydrophobic conditions in otherwise identical nanopores.

## 2. Materials and Methods

### 2.1. Molecular Dynamics Model

Molecular dynamics simulation with atomic-level description is a convenient tool for studying matter under nanoconfinement. To investigate the structure and properties of nanoconfined water, as well as the influence of wall–water electrostatic interaction on water behavior under high-pressure conditions, two molecular dynamics models of a slit-shaped nanopore filled with water were developed. Each model consisted of an approximately 2 nm thick layer of water confined by two parallel metal hydroxide nanosheets located in the *x–y* plane (Figure 1). Periodic boundary conditions (PBCs) were applied along the *x* and *y* axes.

In the first model, the nanopore walls are representative of a class of naturally occurring metal hydroxides Me^(II)^(OH)_2_. The structural parameters of divalent iron hydroxide Fe(OH)_2_ with near-medial lattice constant *a* [56] were chosen as representative parameters for layered hydroxides of divalent metals within the trigonal crystal system (Appendix A).

The second model is equivalent to the first one, except for modified partial atomic charges (PACs), whose values were reduced by a factor of 10 compared to the first model (Appendix A). Henceforth, we will refer to the first model as the “Naturally charged host model” and the second model as the “Low-charged host model”. It should be noted that both models have the overall composition 36[2Me^(II)^(OH)_2_·7H_2_O], and that there is no net electric charge. In other words, we studied two “host-guest” systems with the same atomic structure of the nanopore walls and the same number of “guest” water molecules. The only difference was that the confining walls were hydrophilic in the first model, and hydrophobic in the second.

The crystal structure obtained for deuterated iron(II) hydroxide Fe[O(H_0.86_D_0.14_)]_2_ by Parise et al. [58] was used for the nanosheet wall model. This material was chosen because it has a crystal structure similar to many brucite-like metal hydroxides (such as Ni(OH)_2_, Mg(OH)_2_, Co(OH)_2_, Zn(OH)_2_, Mn(OH)_2_, Cd(OH)_2_, Ca(OH)_2_), and has an intermediate lattice constant among them (see comparison in Appendix A).

The force field parameters that describe the nanopore walls (including the PACs, Lennard-Jones, and covalent bond parameters) were assigned in the naturally charged host model in accordance with the CLAYFF force field [59]. The only exception was that the Lennard-Jones parameters for hydrogen atoms were parameterized as described in [60], with non-zero values *r*_0_ = 0.449 Å and ε = 0.1925 kJ/mol [61], aiming for better compatibility with the TIP3P water model. Partial atomic charges of walls in the low-charged host model system were 10 times lower than in the first model. The PAC values used in the present study are listed in Appendix A.

Both models contain a constant number of water molecules (N_w_ = 7 × 36 = 252) confined between two parallel nanosheets of metal hydroxide. Each nanosheet contains 36 Me^(II)^(OH)_2_ units. The entire system can therefore be described as 36[2Me^(II)^(OH)_2_·7H_2_O].

The TIP3P model was used to parameterize the water molecules [62]. The covalent bonds and angles of the model are described in Appendix A.

The dimensions of the simulation box—X = 16.965 Å and Y = 19.590 Å—remained unchanged during the runs. The size of the system along the *z* axis and, therefore, the width of the nanopore, was self-adjusting depending on external load. The simulation for each applied pressure was 3 ns long.

Larger systems were additionally prepared by replication of the base models twice along both the *x* and *y* axes. The dimensions of the enlarged system were X = 33.93 Å, Y = 39.18 Å, with an overall formula 144[2Me^(II)^(OH)_2_·7H_2_O]. These models contained 1008 water molecules. They were used in the series of additional verifying simulations near the phase transformation points. The duration of each additional simulation was 5 ns.

The cutoff distance for pairwise interactions was 10 Å, with switching to a smooth damping function starting at 8 Å. The long-range term of the electrostatic interaction was evaluated with the PPPM (particle–particle–particle–mesh) method, with a relative error of 10^−4^ [63]. Temperature was constant and equal to 310 K in all simulations. The temperature was maintained by a Nose–Hoover thermostat [64]. The time step was 1 fs.

The traditional approach in molecular dynamics to the implementation of confinement of matter is the use of rigid [65] and elastic [66] walls. We used the rigid wall approximation, with all heavy (non-hydrogen) atoms of the Me^(II)^(OH)_2_ being part of a single rigid body (one rigid body per nanosheet). Thus, the metal and oxygen atoms of the walls were able to move only in the vertical direction *z*. Surface hydrogen atoms, being covalently bonded with oxygen atoms, had no constraints applied.

The model samples were subjected to uniaxial compressional loading along the *z*-axis (see Figure 1). The coordinates of the heavy atoms of the lower wall were fixed during simulation, whereas the upper wall behaved as a piston, providing the specified load on the confined water along the *z* axis (Figure 1). The load was applied by an external force -*f*_z_, which was uniformly distributed over all non-hydrogen atoms of the upper wall piston. The force was directed downward; its magnitude was calculated as the product of the specified pressure and the area of the wall–piston surface. The external (applied) pressure *p* varied in the range of 0.1–10 GPa. This range particularly corresponds to tectonic plates at the characteristic depth interval from 3 km (upper crust) to 300 km (upper mantle) [20], or to highly stressed contact spots in heavily loaded tribological couples [67].

Using the described models, a comprehensive analysis of nanoconfined water structure and properties at different pressures was carried out. The analysis covered:Water density and compressibility;Spatial arrangement of water molecules, including the radial distribution functions (RDFs);The number of hydrogen bonds;The self-diffusion coefficient.

### 2.2. Advantages of the Model

As previously mentioned, the choice of the Fe(OH)_2_ structure with a lattice constant of *a* = 3.265 Å [58] as the basis of the model allowed us to extend the results of the study to the general class of brucite-like layered hydroxides of divalent metals, or at least to those of Ni (3.117 Å), Mg (3.147 Å), Co (3.173 Å), Zn (3.194 Å), Mn (3.316 Å) and, probably, Cd (3.499 Å), due to the similar values of their lattice constants [44].

### 2.3. Assumptions of the Model

We assumed that the utilized force field parameters were sufficiently accurate to model the studied systems at high pressures, and in particular that the TIP3P model was suitable for describing water behavior up to a pressure of 10 GPa

We also assumed that the mineral did not change its structure under loading and, therefore, that the rigid model was applicable (except for surface hydrogen atoms). This assumption is partially supported by data obtained by Hermann and Mookherjee [68]. According to a computationally derived thermodynamic phase diagram for brucite at ~300 K [68], the brucite Mg(OH)_2_ retains its layered structure up to a pressure of ~20 GPa. The rigid model of the wall structure is consistent with these data.

### 2.4. Limitations of the Model

The modeled systems had a fixed ratio of water molecules N_w_ to the number of metal atoms (7:2), and only the case of constant dimensions along the *x* and *y* axes was considered.

## 3. Results

### 3.1. Structure and Properties of Nanoconfined Water

The simulations revealed two solid high-pressure phases of water confined between naturally charged Me^(II)^(OH)_2_ sheets (in addition to the low-pressure liquid phase). These phases can be clearly seen in all numerically obtained dependencies (see Figure 2, Figure 3, Figure 4 and Figure 5a,c). Two phase transitions at *p*_1_~3.1 GPa and *p*_2_~6.7 GPa were determined within the studied pressure range for the base model 36[2Me^(II)^(OH)_2_·7H_2_O] (Figure 2a, black curves). More accurate estimates were obtained using the fourfold-enlarged model 144[2Me^(II)^(OH)_2_·7H_2_O] in prolonged simulations (Figure 4, red curves in the left and central panels): *p*_1_~3.3–3.4 GPa, and *p*_2_~6.7–7.1 GPa.

In the first phase transition at the pressure *p*_1_, the random arrangement of water molecules became ordered (Figure 3a), and the diffusion coefficient of water decreased by three orders of magnitude (Figure 5c, solid black curve) from *D*~3.24 × 10^−10^ m^2^/s to 4.33 × 10^−13^ m^2^/s. It is worth mentioning here that our estimate of the self-diffusion coefficient at *p* = 0.1 GPa for nanoconfined water was 3.1 × 10^−9^ m^2^/s, which is of the same order of magnitude as that of bulk water *D*_bulk_~2.299 × 10^−9^ m^2^/s [69].

According to generally accepted data, ordinary water in the bulk state crystallizes into ice VI [70] at a phase transition pressure of about 1.0–1.17 GPa (at temperatures of 300–310 K) (see Appendix A). However, experimental evidence of the possibility of metastable supercompressed bulk liquid water at higher pressures has also been reported [71]. Dolan et al. [72] were the first to show the formation of cubic ice (ice VII) under shock compression with GPa peak pressures. They particularly showed that the solid phase becomes more stable than the liquid phase at applied pressures above 2 GPa, and that the phase transformation takes place within nanoseconds instead of seconds under ambient conditions. Later, Dolan et al. [73] demonstrated a practical limit for the metastable liquid phase. They showed that bulk water under isentropic shock compression can homogeneously transform into a high-pressure phase (ice VII) at a pressure of 7 GPa (a practical limit for the metastable liquid phase) [73,74,75]. The abovementioned values 1 GPa and 7 GPa bound the pressure range in which crystallization of water is possible at ambient or moderately elevated temperatures. The specific value of crystallization pressure depends (at least for microscale, and even more so for nanoscale volumes of water) on interface hydrophilicity—that is, on hydrogen bonding between the wall surface and water [73,74].

Our results show that nanoconfinement by naturally charged Me^(II)^(OH)_2_ sheets provides an upward shift of the water crystallization pressure by about ~1 GPa in comparison with the abovementioned lower pressure limit for ice VII.

The density of liquid water nanoconfined between naturally charged nanosheets rises from ρ = 1210 kg/m^3^ at 0.5 GPa to ρ = 1540 kg/m^3^ at *p* = 3 GPa (Figure 2a, solid black curve). A rough estimation of the compressibility β of nanoconfined water was obtained using the derivative of the measured density–pressure dependence ρ(*p*) (or, rather, the inverse volume). This result is depicted by a dashed black curve in Figure 2a. Within the pressure range from 0.5 GPa to *p*_1_, the compressibility β decreases from 2 × 10^−10^ to 6 × 10^−11^ Pa^–1^. It should be noted that the isothermal compressibility of bulk water under normal conditions is 4.599 × 10^−10^ Pa^−1^ [76].

The phase transition at *p* = *p*_1_ is accompanied by a stepwise increase in density from ~1540 kg/m^3^ to ~1630 kg/m^3^ (Figure 2a, solid black curve), and by a jump in compressibility (β reaches its peak value at *p*_1_). Common neighbor analysis (CNA) [77] and adaptive CNA [78] for the positions of water oxygen atoms show that the spatial arrangement of water molecules at *p* > *p*_1_ corresponds to the hexagonal close-packed (HCP) type of lattice (Figure 3a). We will henceforth refer to this phase of nanoconfined water as HCP-like. Note that the hydrogen atoms are not included in the structure determination, so this lattice is not a conventional HCP. Radial distribution functions for O-O and O-H pairs of the studied systems at different pressures can be found in Appendix A. Comparing the RDFs obtained at different pressures, one can see the absence of sharp peaks (except the first) in the O-O curves at 2 GPa (red curves), as well as the series of sharp peaks at *p* = 4, 6, and 8 GPa (yellow, green, and blue curves, respectively). This series of peaks is typical for systems with an ordered periodic structure.

It is interesting to note that the point *p*_1_ almost coincides with the pressure 2.7 GPa, at which a pressure-induced water insertion into kaolinite Al_2_Si_2_O_5_(OH)_4_·3H_2_O (a layered hydroxide) is experimentally observed [20]. In this context, it is important that the formation of such a superhydrated phase of mineral may be related to the abrupt reversible increase in compressibility and the transition to a denser molecular packing of the nanoconfined water observed in our simulations (Figure 2a).

The HCP-like phase of nanoconfined water exists in the pressure interval from *p*_1_ to *p*_2_~6.7 GPa (red-shaded region in Figure 2, Figure 3a and Figure 5a,c). According to MD results, water density increases from ρ = 1630 to ρ = 1760 kg/m^3^ with the increase in pressure from *p*_1_ to *p*_2_, while uniaxial compressibility gradually decreases from β = 6 × 10^−11^ Pa^−1^ to β = 2 × 10^−11^ Pa^−1^ (Figure 2a, dashed black curve).

At *p* = *p*_2_, the second phase transition is encountered. Based on CAN, we established a phase transformation from an HCP-like to an FCC-like (face-centered cubic) structure at this pressure. Molecular packing became denser, and the density ρ abruptly increased from 1760 to 1800 kg/m^3^ (Figure 2a, solid black curve). This was also accompanied by a jump in the compressibility β. A further increase in external loading up to 10 GPa led to the densification of this FCC-like phase of nanoconfined water to about 1900 kg/m^3^, without any other structural transformations.

A similar study for the second model (nanoconfined water between low-charged walls) showed that the sequence of changes in the water structure qualitatively differs from the sequence described above for the naturally charged mineral. Water remained in a liquid state up to approximately the same pressure ~3.2 GPa (red solid curves in Figure 2a and Figure 4, right panel) as in the case of the first model (Figure 2a, black curve; Figure 4, left panel). The phase transformation was also manifested in the self-diffusion curve (Figure 5c). However, unlike the case of naturally charged (hydrophilic) walls, water in the nanopores of the low-charged (hydrophobic) host crystallized directly into the FCC-like lattice (Figure 3b). This was accompanied by an abrupt change in the density of nanoconfined water from ρ = 1500 kg/m^3^ to ρ = 1570 kg/m^3^ (Figure 2a), as well as in the jump-like behavior of the β(*p*) curve. Our estimate of water density near the point of crystallization—ρ = 1500 kg/m^3^ at T = 310 K—is similar to the simulation results by Han et al. [79] for water in a quasi-two-dimensional hydrophobic nanopore slit, who showed that water crystallizes at a density of ~1460–1470 kg/m^3^ (T~300 K) into ice with a rhombic formation.

The FCC-like packing of water molecules in the hydrophobic model remains stable at the upper limit *p* = 10 GPa of the investigated pressure range. Note that the density of water at 10 GPa reaches ρ = 1870 kg/m^3^. This is about 2% lower than in the case of confinement by hydrophilic walls.

It is interesting to compare the obtained density–pressure dependencies (Figure 2a) with the theoretical and experimental dependencies for ice VII in the bulk state. Myint et al. [80] developed an equation of state for ice VII, and presented the corresponding theoretical curve ρ(*p*). Comparison of this curve with our dependence shows that theoretically predicted ice VII density values correlate quite well with our values for FCC-like ice in nanopores with hydrophobic walls at pressures up to 7–8 GPa. At higher pressures, the slope of the theoretical curve ρ(*p*) for ice VII becomes flatter than in our case. However, even at a pressure of 10 GPa, the difference in density values in the Mie–Gruneisen equation of state and in the MD model of water in hydrophobic confinement does not exceed 2%. We also note that even this difference is smaller than the difference between the theoretical curve and the experimental data (X-ray diffraction) for ice VII [81]. This raises the question of a possible ambiguity in the interpretation of the crystal lattice of high-pressure ice VII (which is conventionally referred to as the BCC lattice, while the real structure is more complex [82]).

The hydrophilicity of the first model and the hydrophobicity of the second one are also reflected in the water density profile across the nanopore (Figure 2b). The first layer of hydrogen atoms near the hydrophilic walls lies closer to the surface than the first oxygen layer, due to non-covalent bonding with the surface (Figure 2b, green triangular markers). In the case of hydrophobic walls, the position of the first layer of water hydrogen coincides with the first layer of water oxygen atoms.

In order to improve our estimates of the phase transformation pressures, the simulations were repeated with fourfold-enlarged models and a longer runtime (5 ns) near points *p*_1_ and *p*_2_ for the naturally charged mineral model, and near point *p*_1_ for the low-charged mineral model. The results of these MD simulations are depicted in Figure 4. As can be seen from the ρ(*p*) dependencies (red curves in Figure 4) for the naturally charged system, the more accurate value of crystallization pressure is *p*_1_~3.3–3.4 GPa, and the point of HCP-to-FCC transformation is *p*_2_~6.7–7.1 GPa. The more accurate estimate of the crystallization pressure of water in the low-charged model is *p*_1_~3.2–3.3 GPa.

### 3.2. Hydrogen Bonding

A representative quantitative characteristic that reflects the structure and the state of water is the number and topology of hydrogen bonds (H-bonds). Estimates of the average number of water–water H-bonds per molecule for both models are shown in Figure 5a as functions of applied pressure. To estimate the number of H-bonds, we adopted a simple geometric criterion implemented in the MDAnalysis [83,84] module for Python. It was assumed that an H-bond exists if the distance between the donor and acceptor O_acc_–O_don_ is equal to or less than 3.0 Å, with an angle O_acc_–H–O_don_ ≥ 140°. The effect of PBCs was taken into account. The analysis of the curves in Figure 5a shows that the change in the average number of water–water hydrogen bonds per molecule correlates with structural transformations of nanoconfined water.

In a naturally charged nanopore, there are hydrogen bonds between water molecules and the walls (Figure 3a and Appendix A). The number of water–wall H-bonds per surface OH- group in a naturally charged nanopore is shown in Appendix A as a function of pressure. Surface OH- groups act both as proton donors and as acceptors (Appendix A). This confirms the hydrophilic properties of the naturally charged host, and is consistent with experimental knowledge of the considered mineral [85]. We should note that in the pressure interval between *p*_1_ and *p*_2_ (HCP-like crystal structure of water), the water–water H-bonds lie predominantly in the *x–z* plane, and form an angle θ~60° with the plane of the mineral surface (Figure 3a and Appendix A). On the other hand, the surface of the low-charged host mineral behaves rather hydrophobically. This is confirmed by the predominant parallel orientation of water molecules to the surface of the mineral (Figure 5d and Appendix A), as well as by the predominant formation of water–water H-bonds in this plane, θ~0° (Figure 3b). The latter is also supported by the observation that both oxygen and hydrogen atoms of the water cluster in the same *z*-layer (Figure 2b, row “Low-charged walls”).

At pressures corresponding to the liquid state of nanoconfined water (at *p* < *p*_1_), the average number of H-bonds gradually increases for both considered systems. When passing through the crystallization point (near *p*_1_), both systems exhibit a stepwise increase in the average number of hydrogen bonds per molecule (Figure 5a). At pressures above *p*_1_ (nanoconfined water in the crystalline state), the nature of the change in the number of H-bonds in the simulated systems with increasing *p* differs significantly.

In the case of water confined between the naturally charged walls, the dependence of the average number of H-bonds on pressure within the pressure range *p*_1_
*< p < p*_2_ (ice with HCP-like structure) has the form of a relatively high plateau (Figure 5a, black curve). At the point of phase transition from HCP-like to FCC-like structure (at *p*_2_), the number of H-bonds per molecule drops back to a level almost equal to that near *p*_1_. In the case of low-charged (hydrophobic) walls, the estimate of the number of H-bonds in nanoconfined water at *p > p*_1_ (ice with an FCC-like structure) fluctuates around a constant value (Figure 5a, red curve). However, this value is always lower than in the case of naturally charged walls.

The described difference in the pressure dependencies of the average number of H-bonds between hydrophilic and hydrophobic nanopore walls is a consequence of the difference in the preferential orientation of water molecules in the near-surface layer. The oriented structure of hydrogen bonds in water molecules near the mineral surface affects the structure of hydrogen bonds in the entire volume of water in slit-shaped nanopores at GPa pressures. In the naturally charged nanopores, the H-bonds between the wall surface atoms and near-surface water molecules determine the transition of confined water into the HCP-like crystal structure at the point *p*_1_. With further increase in the applied pressure, due to the decrease in the intermolecular distance in the deformed HCP-like ice, the contribution of the repulsive van der Waals component of intermolecular interaction becomes significant. As the applied pressure approaches *p*_2_, the contribution of van der Waals force becomes predominant over the contribution of hydrogen bonding. This determines the HCP-to-FCC phase transformation of the nanoconfined water in a naturally charged nanopore at *p*_2_. In the low-charged nanopores, there are no water–wall H-bonds, since the artificial surface is hydrophobic. Hydrophobic walls cause near-surface water molecules to orient themselves mainly parallel to the plane of the slit, and the crystallization of liquid water in the nanopore into the FCC-like phase occurs directly at *p*_1_.

Summarizing the above, the surface polarity of the nanopore walls controls the interaction of the first water layer with the nanopore surface. Under conditions of nanoconfinement and high pressure, this determines both the preferential orientation of water molecules and the structure of the H-bonds network in the entire nanopore volume, thereby affecting the properties of the nanoconfined water. Ultimately, this also affects the effective physical properties of the water-saturated mineral.

### 3.3. Mechanical Response of Confined Water to Compression

Numerically predicted stress–strain curves (response functions) for the compression testing of water nanoconfined between hydrophilic and hydrophobic walls are shown in Figure 5b. Note that due to the fixed dimensions of the system in the *x–y* transverse plane, the axial strain of the water layer is numerically equal to its volumetric strain, and the axial stress (normal pressure) can be considered as a rough estimation of hydrostatic pressure. Therefore, the above diagram can be considered as an alternative way of representing the density–pressure dependence. The only liquid-to-solid transformation in the low-charged nanopores and the two phase transformations (liquid-to-HCP and HCP-to-FCC) in the naturally charged nanopores appear on the stress–strain curves in the form of near-horizontal segments: I_1_ and II_1_—for naturally charged walls, and I_2_—for low-charged walls. In these segments, the compression of water in the nanopores occurs almost without an increase in the opposing force, i.e., the instantaneous values of the water compressibility
$$\beta =-\frac{1}{V}\frac{dV}{dp}$$(where V is pore volume) experience sharp jumps (Figure 2a, dashed curves). In the pressure region near *p*_1_, compressibility increases by a factor of 4–5. Even more impressive is the order-of-magnitude jump in water compressibility at the HCP-to-FCC transition in hydrophilic nanopores (near *p*_2_). It is clear that the jumps in the compressibility of interstitial water lead to corresponding abrupt fluctuations in the effective value of the compressibility of the whole nanoporous mineral. The amplitude of these fluctuations is determined by the volume fraction of watered nanopores and the specific surface area of the nanopores. We should also note that, although the σ–ε curves (Figure 5b) for both models are fairly close in general, the different positions and the different numbers of horizontal sections (two in the first model and one in the second model) may cause a significant difference in the mechanical behavior of minerals with different polarity of the wall surface at high pressures.

## 4. Discussion of Geoscience-Related and Friction-Related Results

The influence of the surface energy of confining walls on the pressure of dynamic phase transition of microscopically thin samples of water to a high-pressure crystalline phase has been studied by various researchers [73,74,86]. The most significant result is the demonstrated important role of hydrophilicity/hydrophobicity of the surface. If the confining surface has functional groups that provide sufficient surface energy to form sufficiently strong hydrogen bonds with water, this promotes heterogeneous nucleation of water crystallization (from the surface–water interface inwards) at significantly lower pressures as compared with walls characterized by low surface energy. Since surface energy cannot be independently varied in experiments, the authors of these studies used various wall materials (e.g., silica, aluminum, sapphire) with different chemical compositions and structures. Therefore, the observed strong differences in the values of water crystallization pressure are determined not only by different surface polarity, but also to a large extent by the difference in the crystal structure of wall materials. In this paper, we considered for the first time the effect of wall surface polarity as a separate, independent factor. We found for the first time that this can determine not only the water crystallization pressure, but also the formation of an atypical intermediate high-pressure phase (HCP) that is stable in a limited pressure range.

A comparative analysis of both models shows that the physicomechanical properties of nanoconfined water in the liquid phase are determined not only by the magnitude of ambient pressure, but also by the electric charge distribution on the mineral surface. In particular, the water density in nanopores of the low-charged host mineral is 5% lower than in the naturally charged mineral. At the same time, liquid water in the low-charged mineral has a slightly higher compressibility (Figure 2a, dashed curves). Such differences in the density and compressibility of water can have a significant effect on the stress–strain state of water-saturated minerals. Indeed, the 5% difference in the density of the water determines the corresponding difference in pore volume at a given ambient pressure. This, in turn, determines the difference in the density of water-saturated minerals with similar thermal and mechanical characteristics of the solid matrix but with significantly different surface polarity of the pore walls. Under the conditions of the lower layers of the Earth’s crust and the upper layers of the asthenosphere, such a difference in the density values of water-saturated nanoporous rocks will lead to a difference in their stress state. In the simplest approximation, this difference can be estimated using Biot’s model of linear poroelasticity [87,88]. Within the framework of this model, the contribution of the pore pressure to the magnitude of the volumetric stress in the water-saturated material depends linearly on the porosity and the compressibility of the fluid. Therefore, the abovementioned differences in the density and compressibility of confined water will lead to a difference of up to several percent in the magnitude of volumetric stresses in the rock matrix.

The influence of wall surface polarity on the structure and properties of nanoconfined water becomes more essential at pressures above the crystallization point *p*_1_. The electrostatic interaction of the surface of nanopores with adjacent layers of water in nanopores determines the crystalline phase of water in the pressure range ~3–7 GPa, which roughly corresponds to the depth interval of 80–200 km (the existence of fluid-rich locations in the upper transition zone and deep mantle of the Earth are confirmed by recent studies [89]). In this depth interval, nanoconfined ice may be in the HCP-like phase (in a hydrophilic host mineral) or the FCC-like phase (in a hydrophobic host mineral). Differences in the crystal structure of interstitial ice will, in turn, influence the effective thermal and mechanical characteristics of nanoporous minerals.

The demonstrated existence of an “intermediate” HCP phase of nanoconfined ice in minerals with hydrophilic surfaces is important for understanding and interpreting various aspects of the subduction process. The abrupt change in compressibility, related to the HCP-like–FCC-like transition at *p*_2_, may induce an abrupt change in the effective mechanical properties of water-saturated minerals of a subducting slab at a depth of about ~200 km. Such a structural transformation of ice and abrupt changes in properties should not be found in minerals with low (partial) electric charges on the pore surfaces. This difference can thus partially determine the behavior of hydrophilic and hydrophobic rocks at the subduction interface.

The difference in the crystalline phases of water nanofilms confined by hydrophilic and hydrophobic surfaces in the pressure range 3–7 GPa can have a significant effect on the dynamics of friction. It is known that if the width of the lubricant film is of the order of a few molecular layers, then at pressures sufficient to solidify the lubricant, the friction of the surfaces is realized in the stick–slip mode [26,27,90]. The stick–slip mechanism in this case is the alternation of the melting (slip phase) and freezing (stick phase) processes. It is clear that accompanying dynamic effects at the stages of relative acceleration and deceleration of surfaces (e.g., the energy of elastic waves, including seismic waves; dilatancy/compaction, etc.) are determined by the crystalline phase of the lubricant. The results of this work, which demonstrate different preferred crystalline phases of water nanoconfined by hydrophilic and hydrophobic surfaces, suggest that these effects would be different for minerals with different degrees of hydrophilicity. For example, [55] reports the results of an MD study of the friction of hydrophilic (crystalline mica) and hydrophobic (graphene) surfaces separated by a nanofilm of water (<1 nm). It was shown that a decrease in the hydrophilicity of surfaces is accompanied by a change in the friction regime from stick–slip to quasi-viscous (“smooth sliding”). These results, although obtained at much lower values of the normal pressure, give grounds to assume that similar trends (i.e., smoothing of the friction regime) should also take place at GPa pressures. Thus, understanding the regularities of friction of surfaces separated by nanofilms of water helps to better understand the behavioral features of subducting slabs at depths from tens to hundreds of kilometers. These regularities should be taken into consideration when constructing large-scale models of subduction zones.

The hydrophilicity/hydrophobicity of confining surfaces can have a significant impact on the dynamics of friction in technical tribosystems in the presence of water as a lubricant. Due to the multiscale roughness of real surfaces, distribution of local contact pressure is extremely inhomogeneous. Local contact pressures may reach gigapascals in submicron and nanoscopic contact spots (the latter are especially important for ceramic materials [91]). It seems that the abovementioned effect of nanoscopic water layers on the frictional behavior of such contact spots can potentially be taken into account to control the friction coefficient and the friction mode of heavily loaded friction pairs by developing surface layers with specified characteristics of surface polarity.

Finally, we would like to make a few comments about the term “confining pressure”, which is often used in papers devoted to the study of nanoscopic volumes of water. Recent experimental and theoretical results provide contradictory data on phase transformations of water at high (GPa) pressures. There is a well-known and thoroughly studied effect of high-pressure-induced local melting of ice, and refreezing once the pressure is lifted [92]. This effect explains the melting of 2D ice (a few layers of water molecules) between solid surfaces when the applied normal load reaches a specific critical value. The critical melting pressure *P_M_* is of the order of several GPa [46]. When the temperature is increased, the melting pressure decreases (note that the value *P_M_*~3.5 GPa at *T* = 310 K reported in [46] is close to the crystallization point of nanoconfined water *p*_1_~3.3–3.4 GPa in our study). Pressure-induced ice melting is explained by concurrent shortening and stiffening of hydrogen bonds O:H, and by elongation and softening of covalent bonds O-H (these changes lead to O:H-O bond energy loss) [93]. Such a mechanism does not, however, explain the results of other studies, including a recent ab initio study [94]. This study showed stable double-layer ice and only solid-phase structural transformations at GPa confining pressures. Moreover, our present study for six layers of water molecules also showed no pressure-induced melting in the pressure range from 3 to 10 GPa. We believe that the qualitative difference in the observed behavior of nanoconfined water is due to a difference in the confinement mode. In the case of water between two sheets, pressure is applied along one axis, and the water is relatively free in the transverse plane (1D confinement). This facilitates both the reorientation of molecules and structural rearrangements accompanied by loss of order. The 3D confinement considered in this paper and in [94] (in both studies, the confinement in the transverse plane was implemented using periodic boundary conditions) contributes to the stabilization of the ordered structure. We suppose that the ratio of deformations of O:H and O-H bonds under 3D confinement is significantly different from that under 1D confinement. This issue deserves separate study and discussion.

The foregoing analysis indicates that the behavior of nanoscopic water layers in natural and technical tribological systems is determined not simply by the contact pressure, but also by the difference between pressures applied along mutually perpendicular axes (the term differential stresses is used in rock mechanics). In the case of low differential stresses (which correspond to water in nanopores of rocks, or in nanoscopic “depressions” in contact spots of rubbing bodies), the dependencies of the structure and mechanical properties of water on contact pressure largely correspond to those described in this paper. In the case of high differential stresses close to the applied contact pressure (i.e., water between flat areas of the contacting surfaces), a mixed structure of nanoconfined water is expected (i.e., adjacent ice and quasi-liquid regions [46]). The resistance of such a structure to shear loading is rather viscous (i.e., smooth sliding), regardless of the hydrophilicity of the confining surfaces. The discussed diverse effects of nanoscopic water layers on friction in contact spots should be taken into account when interpreting the results of studies of lubricated friction with water as a lubricant.

## 5. Conclusions

To the best of our knowledge, this work is the first study of the structure and properties of water nanoconfined between two sheets of naturally occurring layered brucite-like minerals. The studied systems were examined under high-pressure conditions in the range of 0.1 to 10 GPa. The influence of the surface polarity of nanopore walls on the structure and the properties of nanoconfined water was elucidated. Molecular dynamics results revealed two pressure-driven phase transformations of nanoconfined water in an approximately 2 nm thick slit-shaped nanopore with hydrophilic walls: at *p*_1_~3.3–3.4 GPa (pressure-induced crystallization into HCP-like lattice), and at *p*_2_~6.7–7.1 GPa (transformation from HCP-like to a dense FCC-like crystal). Only one phase transition (from liquid to FCC-like crystal) was observed for water confined in the artificial hydrophobic nanopores with the same atomic structure but lower surface polarity. It was shown that, under conditions of nanoconfinement and high pressure, the interaction of the first layer of water with the surface of the nanopore controls the predominant orientation of water molecules and, thereby, the structure of the hydrogen bond network in the whole nanopore volume, thus determining the phase state and the properties of nanoconfined water.

## Figures and Tables

**Figure 1 materials-15-03043-f001:**
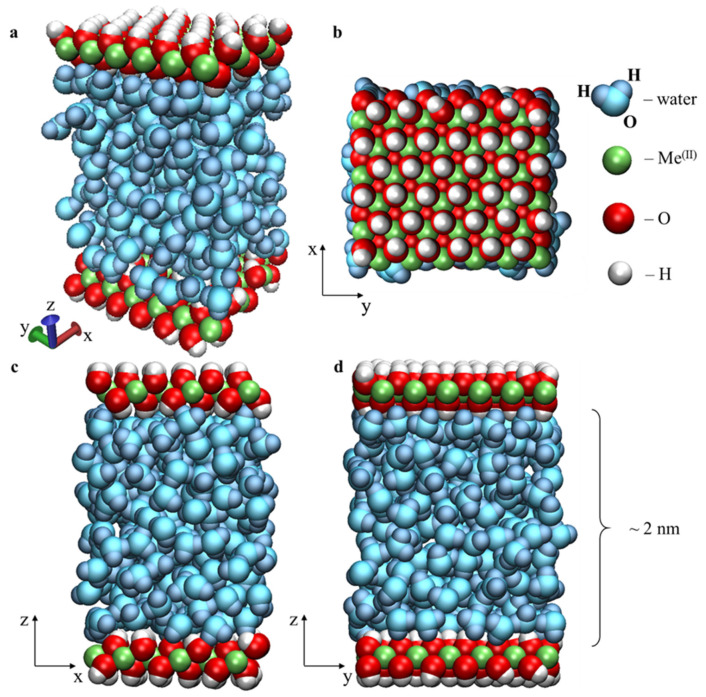
Initial configuration of the modelled system’s periodic boxes 36[2Me^(II)^(OH)_2_·7H_2_O] equilibrated at the pressure 0.1 GPa: perspective view (**a**), top view (**b**), *x–z* view (**c**), and *y–z* views (**d**).

**Figure 2 materials-15-03043-f002:**
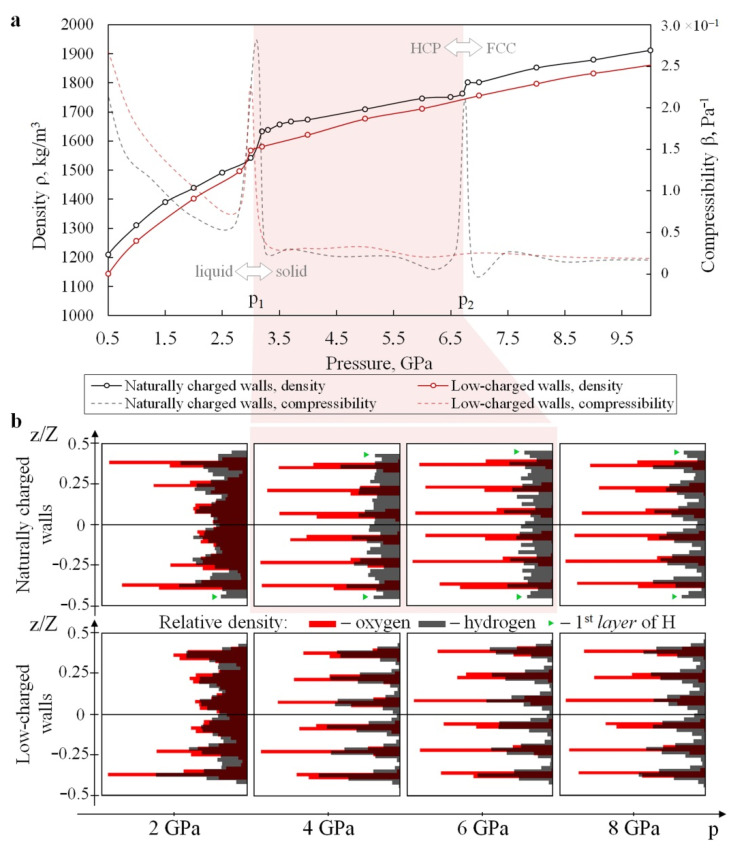
Properties and phases of nanoconfined water depending on applied pressure and wall surface polarity. Results were obtained for the base models of 36[2Me^(II)^(OH)_2_·7H_2_O]: (**a**) Numerically obtained density–pressure diagram for confined water in both studied systems in the pressure range 0.5–10 GPa. The shaded region indicates the crystalline phase with an HCP-like structure of water in the naturally charged host model; the region on the left corresponds to the liquid phase of confined water; the region on the right corresponds to ice with the high-density FCC-like structure (also in the naturally charged host model). (**b**) One-dimensional (along z) density profiles of water oxygen (red bars) and hydrogen (grey bars) atoms in the naturally charged (**top**) and low-charged (**bottom**) minerals at different pressures. The presence of six water layers is evident. The first layers of hydrogen atoms in the proximity of the hydrophilic walls are indicated by small green triangular markers. In the case of hydrophobic walls, the position of the first layer of H coincides with the first layer of O.

**Figure 3 materials-15-03043-f003:**
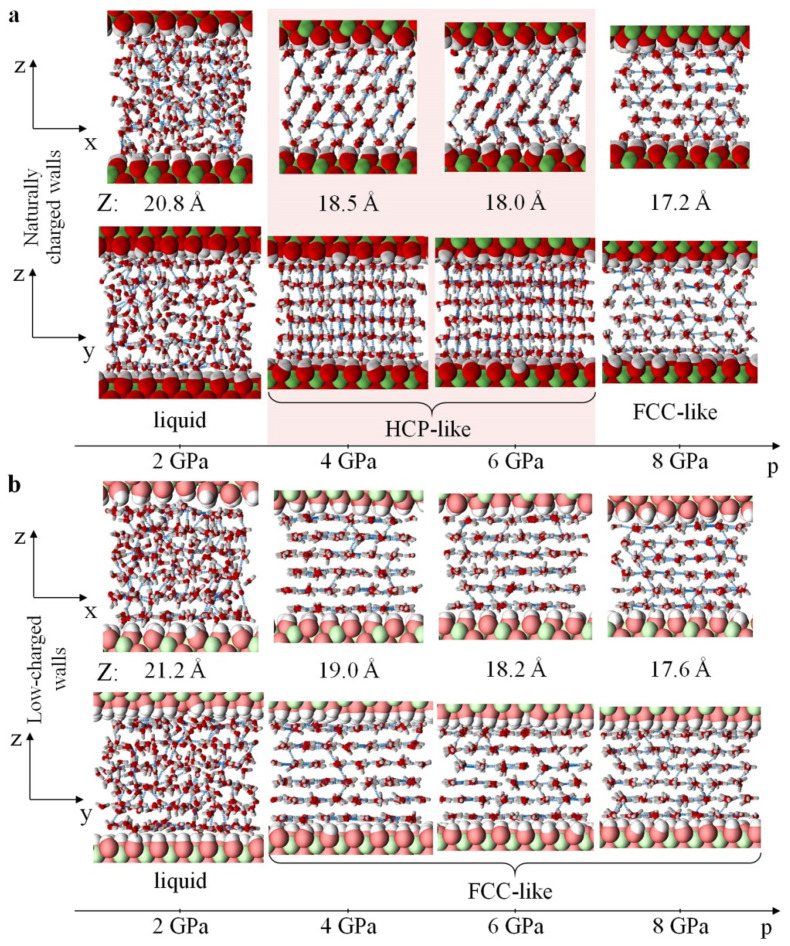
Structure of nanoconfined water depending on pressure and wall surface polarity: Representative views of nanoconfined water in the naturally charged (**a**) and low-charged (**b**) host models in the considered pressure range. Color code: Me^(II)^—green, O—red, H—white, hydrogen bonds—light blue; pale colors of wall atoms indicate 10-fold-reduced partial atomic charges. The results are for the base models of 36[2Me^(II)^(OH)_2_·7H_2_O].

**Figure 4 materials-15-03043-f004:**
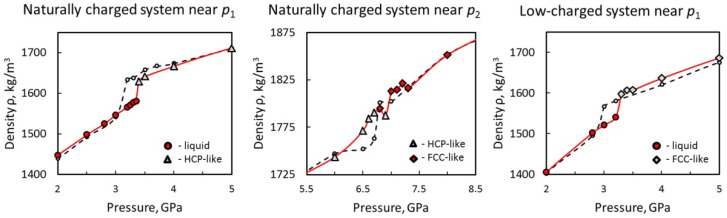
Results of extended (up to 5 ns) simulations for the fourfold-enlarged models of 144[2Me^(II)^(OH)_2_·7H_2_O], containing 1008 water molecules, near the phase transformation points. The estimates of confined water density are consistent with the smaller models, but a slight shift of the phase transition points by about +(0.1–0.3) GPa is observed in the enlarged system. According to these results, for the naturally charged system *p*_1_~3.3–3.4 GPa and *p*_2_~6.7–7.1 GPa, and for the low-charged system *p*_1_~3.2–3.3 GPa. Colors: red solid lines for the enlarged system (red circles represent the liquid phase, white triangles represent the HCP-like phase, and red rhombuses represent the FCC-like phase of the naturally charged system; white rhombuses represent the FCC-like phase of the low-charged system); black dashed lines (small circular markers) for the base model.

**Figure 5 materials-15-03043-f005:**
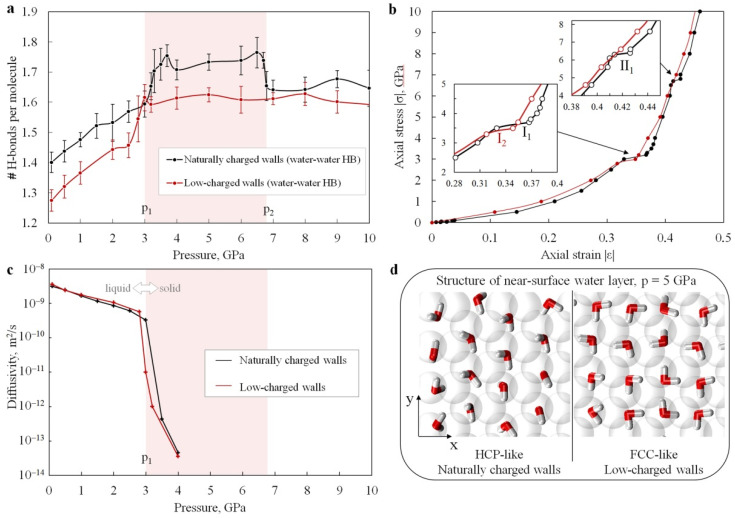
Physical and mechanical properties of nanoconfined water in the systems studied, obtained using the base models of 36[2Me^(II)^(OH)_2_·7H_2_O]: (**a**) Average number of water–water hydrogen bonds per molecule. The red-shaded region represents the HCP-like phase in the naturally charged host model. (**b**) Estimated compression diagrams of water confined in naturally charged (black curve) and low-charged (red curve) minerals. Insets show enlarged diagram fragments for near-horizontal regions of the loading curves. (**c**) Diffusion coefficient of the nanoconfined water as a function of applied pressure, evaluated using mean square displacement (MSD). (**d**) In-plane views of the first near-surface layer of water confined in the nanopores with different wall surface polarity at *p* = 5 GPa. Colors: hydrogen—white, water oxygen—red, mineral oxygen—transparent spheres.

## Data Availability

All necessary data and models will be uploaded to the GitHub repository at https://github.com/AATsukanov/Nanoconfined-TIP3P (accessed on 29 March 2022) and/or https://tsukanov-lab.moy.su (accessed on 29 March 2022). The data that support the findings of this study are also available from the corresponding authors upon request. Four well-known codes/packages were used to produce the results: the “Visual Molecular Dynamics” (VMD) package (available at http://www.ks.uiuc.edu/Research/vmd/, accessed on 29 March 2022) for data preparation and graphical visualization of the results [104]; the CUDA [105] version of the LAMMPS (Large-Scale Atomic/Molecular Massively Parallel Simulator) package for MD simulations (available at http://lammps.sandia.gov, accessed on 29 March 2022) [106,107]; the “MDAnalysis” Python library for hydrogen bond analysis (available at https://www.mdanalysis.org, accessed on 29 March 2022) [83,84]; and OVITO (available at http://www.ovito.org, accessed on 29 March 2022) for conventional and adaptive common neighbor analysis [108].

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
