# Peer review of "Structure, Properties, and Phase Transformations of Water Nanoconfined between Brucite-like Layers: The Role of Wall Surface Polarity"

_materials, 2022, doi:10.3390/ma15093043_

Round 1

Reviewer 1 Report

Referee report Manuscript ID: materials-1682268

Title: Structure, Properties and Phase Transformations of Water Nanoconfined between Brucite-like Layers: Role of Wall Surface Polarity

By Tsukanov, Shilko and Popov

This manuscript reports on an interesting and detailed molecular dynamics simulation study on the structure, properties and phase transitions of water confined between two nearby brucite-like walls. The pressure is varied in range of 0.1 GPa to 10 GPa. By varying the pressure the authors found two phase transitions in their molecular dynamics simulations (at about 3.3 GPa a crystallization from liquid to a hcp structure and at about 7 Gpa a structural hcp to fcc phase transition). The molecular dynamics simulations reveal that the surface polarity of the walls play a pivotal role.

The manuscript has a proper introduction and the story line of the manuscript is easy to follow. Unfortunately, the manuscript contains several textual deficiencies and inaccuracies and therefore I strongly recommend the authors to proofread their manuscript once again. The figures are clear and as far as I can judge the molecular dynamics simulation are carried out in a proper way. The conclusions drawn by the authors are well-supported by their molecular dynamics simulations (The authors do not overinterpret their results ). All in all I believe this is an interesting manuscript that deserves to be published. I have only a few minor remarks:

  • I encourage the authors need to proofread their manuscript as it contains several textual deficiencies and inaccuracies
  • At some locations in the manuscript the authors can make contact with recent experimental work, see for instance pressure-induced melting of ice confined between mica and graphene (see for instance ACS Nano 11, 12723 (2017) and references therein).

In summary, this is an interesting theoretical manuscript on a relevant and timely topic. After the revision the manuscript is in principle suitable for publication.

Author Response

This manuscript reports on an interesting and detailed molecular dynamics simulation study on the structure, properties and phase transitions of water confined between two nearby brucite-like walls. The pressure is varied in range of 0.1 GPa to 10 GPa. By varying the pressure the authors found two phase transitions in their molecular dynamics simulations (at about 3.3 GPa a crystallization from liquid to a hcp structure and at about 7 Gpa a structural hcp to fcc phase transition). The molecular dynamics simulations reveal that the surface polarity of the walls play a pivotal role.

The manuscript has a proper introduction and the story line of the manuscript is easy to follow. Unfortunately, the manuscript contains several textual deficiencies and inaccuracies and therefore I strongly recommend the authors to proofread their manuscript once again. The figures are clear and as far as I can judge the molecular dynamics simulation are carried out in a proper way. The conclusions drawn by the authors are well-supported by their molecular dynamics simulations (The authors do not overinterpret their results ). All in all I believe this is an interesting manuscript that deserves to be published.

Response: We thank the reviewer for the positive evaluation of the paper and valuable comments. We have made changes to the manuscript in accordance with the suggestions of all reviewers. Significant changes are highlighted in yellow in the revised manuscript.

Minor remarks:

1) I encourage the authors need to proofread their manuscript as it contains several textual deficiencies and inaccuracies

Response: We are grateful to the reviewer for attention to the quality of English. We proofread the revised version of the manuscript. The quality of English was thoroughly revised by an English-speaking professional (native speaker of English).

2) At some locations in the manuscript the authors can make contact with recent experimental work, see for instance pressure-induced melting of ice confined between mica and graphene (see for instance ACS Nano 11, 12723 (2017) and references therein).

Response: We are familiar with this very interesting experimental work and similar studies. We thank the reviewer for this reference and for the suggestion to discuss the results of this study in the context of our study. We have mentioned two recent articles (including ACS Nano (2017) 11, 12723) in section “1. Introduction”. Moreover, we briefly discussed the possible reasons for the different behavior of nanoconfined water under high pressure in various researches (two final paragraphs in section “4. Discussion on geoscience-related and friction-related results”).

Reviewer 2 Report

The authors studied nanoconfined water between brucite-like layers. Nanoconfined water was studied under pressures of 0.1 to 10 GPa. The influence of surface polarity was examined. Two pressure-driven phase transformations were found with hydrophilic walls and one in the hydrophobic wall surfaces. The interaction of the first layer of water with the surface determines the phase state.
Methods are appropriate and well described. The results are interesting and novel.
Therefore, in my opinion, this manuscript should be accepted.

Author Response

The authors studied nanoconfined water between brucite-like layers. Nanoconfined water was studied under pressures of 0.1 to 10 GPa. The influence of surface polarity was examined. Two pressure-driven phase transformations were found with hydrophilic walls and one in the hydrophobic wall surfaces. The interaction of the first layer of water with the surface determines the phase state.

Methods are appropriate and well described. The results are interesting and novel.

Therefore, in my opinion, this manuscript should be accepted.

Response: We thank the reviewer for the positive evaluation of the paper and valuable comments. We have made changes to the manuscript in accordance with the suggestions of all reviewers. Significant changes are highlighted in yellow in the revised manuscript.

We proofread the revised version of the manuscript. The quality of English was thoroughly revised by an English-speaking professional (native speaker of English).

Reviewer 3 Report

see attached file

Author Response

In this work, A. A. Tsukanov and co-workers have studied the structure and properties of nanoscale water layers confined between layered metal hydroxide surfaces with brucite-like structure.

Overall, the manuscript appears to be well presented, the introduction provides a detailed background, the methods section should be revisited and part of the results section should be moved to method part (see comments), the results section is well presented, but I had some difficulties in reading, it is long it could be slightly shortened, conclusions are adequate. For these reasons I think that the manuscript deserve to be published in Materials journal after minor revisions reported below.

Response: We thank the reviewer for the positive evaluation of the paper and valuable comments. We have made changes to the manuscript in accordance with the suggestions of all reviewers. Significant changes are highlighted in yellow in the revised manuscript.

We proofread the revised version of the manuscript. The quality of English was thoroughly revised by an English-speaking professional (native speaker of English).

Comments

Introduction

1) Line 32-35:” Solid and liquid phases confined to a volume of several atomic sizes (nanoconfined matter) may significantly or even qualitatively differ from the corresponding bulk phases. This is valid both for particles or droplets of nano size and liquids or solids confined within nanosized pores or gaps.”

Dear authors could you add some references related to this part?

Response: Thank you for this comment. We added four references related to this statement (references 2-5 in the revised manuscript):

Munoz-Santiburcio, D.; Marx, D. Chemistry in nanoconfined water. Chem. Sci. 2017, 8, 3444–3452. https://doi.org/10.1039/c6sc04989c.

Uskov, A.V.; Nefedov, D.Y.; Charnaya, E.V.; Haase, J.; Michel, D.; Kumzerov, Yu.A.; Fokin, A.V.; Bugaev, A.S. Polymorphism of metallic sodium under nanoconfinement. Nano Lett. 2016, 16, 791–794. https://doi.org/10.1021/acs.nanolett.5b04841.

Charnaya, E.V.; Lee, M.K.; Chang, L.J.; Kumzerov, Y.A.; Fokin, A.V.; Samoylovich, M.I.; Bugaev, A. S. Impact of opal nanoconfinement on electronic properties of sodium particles: NMR studies. Phys. Lett. A 2015, 379, 705–709. https://doi.org/10.1016/j.physleta.2014.12.028.

Latysheva, E.N.; Pirozerskii, A.L.; Charnaya, E.V.; Kumzerov, Y.A.; Fokin, A.V.; Nedbai, A.I.; Bugaev, A.S. Polymorphism of Ga-In alloys in nanoconfinement conditions. Phys. Solid State 2015, 57, 131–135. https://doi.org/10.1134/S1063783415010187.

2) Line 35-37: “Under conditions of nanoconfinement many kinetic and thermodynamic processes, including chemical reactions, demonstrate extraordinary behavior not observed in bulk conditions”

Dear authors could you add some references related to this part?

Response: Thank you for this comment. We added three related references (references 6-8 in the revised manuscript):

Huber, P. Soft matter in hard confinement: phase transition thermodynamics, structure, texture, diffusion and flow in nanoporous media. J. Phys. Condens. Matter 2015, 27, 103102. https://doi.org/10.1088/0953-8984/27/10/103102.

Hamilton, B.D.; Ha, J.M.; Hillmyer, M.A.; Ward, M.D. Manipulating crystal growth and polymorphism by confinement in nanoscale crystallization chambers. Acc. Chem. Res. 2012, 45, 414–423. https://doi.org/10.1021/ar200147v.

Yang, Z.C.; Qian, J.S.; Yu, A.Q.; Pan, B.C. Singlet oxygen mediated iron-based Fenton-like catalysis under nanoconfinement. Proc. Natl. Acad. Sci. U. S. A. 2019, 116, 6659–6664. https://doi.org/10.1073/pnas.1819382116.

3) Line 40-42: “Confined ionic liquids form the base for the development of supercapacitors with large capacity and fast charge-discharge capability”

Dear authors, this sentence was not clear for me, please could you modify this sentence?

Response: Thank you for this comment. We modified this sentence:

“Confined ionic liquids are used in promising supercapacitors (electrochemical devices which store energy through reversible adsorption of ionic species) with enhanced capacitance and the capability of fast charging and discharging [12,13].”

4) Line 44-50: “conditions is found everywhere in geological media, thereby making nanoconfinement effects essential for proper understanding of many phenomena in geosciences, including physical and mechanical properties of saturated porous media, formation of super-hydrated phases of minerals, dehydration processes of the subducting slab, deformation and rupture processes of heterogeneous geomaterials, as well as lubrication effects under high-pressure conditions [11,12].

Dear authors, would it be possible to add references for each phenomena in geosciences? (Example: including physical and mechanical properties of saturated porous media [ref], formation of super-hydrated phases of minerals [ref],….).

Response: Thank you for this comment. Several references were added to this sentence:

“…physical and mechanical properties of saturated porous media [18], formation of super-hydrated phases of minerals [19,20], dehydration processes of subducting slabs [20], deformation and rupture of heterogeneous geomaterials [21], as well as lubrication effects under high-pressure conditions [22,23].”

[18] Gor, G.Y.; Gurevich, B. Gassmann theory applies to nanoporous media. Geophys. Res. Lett. 2018, 45(1), 146–155. https://doi.org/10.1002/2017GL075321.

[19] Delville, A. Structure and properties of confined liquids: A molecular model of the clay-water interface. J. Phys. Chem. 1993, 97, 9703–9712. https://doi.org/10.1021/j100140a029.

[20] Hwang, H.; Seoung, D.; Lee, Y.; Liu, Z.; Liermann, H.-P.; Cynn, H.; Kao, C.-C.; Mao, H.-K. A role for subducted super-hydrated kaolinite in Earth’s deep water cycle. Nat. Geosci. 2017, 10, 947–953. https://doi.org/10.1038/s41561-017-0008-1.

[21] Tsukanov, A.; Shilko, E.; Psakhie, S. Structural Transformations of the Nanoconfined Water at High Pressures: A Potential Factor for Dynamic Rupture in the Subduction Zones. In Trigger Effects in Geosystems; Kocharyan, G., Lyakhov, A., Eds.; Springer, Cham, 2019; pp. 297–306.

[22] Popov, V.L. Thermodynamics and kinetics of shear-induced melting of a thin layer of lubricant confined between solids. Tech. Phys. 2001, 46, 605–615. https://doi.org/10.1134/1.1372955.

[23] Huang, X.; Wu, J.; Zhu, Y.; Zhang, Y.; Feng, X.; Lu, X. Flow-resistance analysis of nano-confined fluids inspired from liquid nano-lubrication: A review. Chin. J. Chem. Eng. 2017, 25, 1552–1562. https://doi.org/10.1016/j.cjche.2017.05.005.

5) Line 75-80:” In the confined state, water plays a key role in many practically significant processes, and is important for the design of high-performance fuel cells, self-assembling nanoscale materials and in understanding the functioning of biological molecular machines [39], including selective ion filtration through biological channels, functioning of porins and the change of properties of biological components in soft nanoconfinement [40].”

Dear authors, would it be possible to add references for each significant processes?

Response: Thank you for this comment. We reformulated this sentence and added some references:

“Confined water plays a key role in many practically significant processes: functioning of biological molecular machines [50]; selective ion filtration through biological channels [51]; functioning of porins, and the change of properties of biological components in soft nanoconfinement [52].”

[50] Giovambattista, N.; Rossky, P.J.; Debenedetti, P.G. Computational studies of pressure, temperature, and surface effects on the structure and thermodynamics of confined water. Annu. Rev. Phys. Chem. 2012, 63, 179–200. https://doi.org/10.1146/annurev-physchem-032811-112007.

[51] Åqvist, J.; Luzhkov, V. Ion permeation mechanism of the potassium channel. Nature 2000, 404, 881–884. https://doi.org/10.1038/35009114.

[52] Manni, L.S.; Assenza, S.; Duss, M.; Vallooran, J.J.; Juranyi, F.; Jurt, S.; Zarbe, O.; Landau, E.M.; Mezzenga, R. Soft biomimetic nanoconfinement promotes amorphous water over ice. Nat. Nanotechnol. 2019, 14, 609–615. https://doi.org/10.1038/s41565-019-0415-0.

6) Line 91-92: “This class of minerals tends to form stacked nanolayers with hydrophilic interlayer regions.

Dear authors could you add reference?

Response: Thank you for this comment. Here we mentioned that brucite-like hydroxides of divalent metal have a crystal structure like sandwiched stack of nanolayers. Since this statement is not important for the article, we have removed it.

7) Line 93-94:”we artificially lowered the surface polarity of the chosen Me(II)(OH)2 material without changing its atomic structure.”

Dear Authors, I believe that this sentence should be moved in Method. Materials and Methods

Response: Thank you for this comment. Since section “2. Materials and Methods” contains more detailed information on this subject, we removed this sentence from the article.

8) Line 103-104:” We used the rigid walls approximation, in which all heavy (non-hydrogen) atoms of the Me(II)(OH)2 were represented as parts of a single rigid body (one rigid body for each 104 nanosheet).”

In this sentence authors report the use of rigid walls approximation, is this approximation based on the use of a constraint between the atoms of the surface? Could the sentence be clarified?

Response: Thank you for this comment. We added the sentence for clarification as follows:

“We used the rigid wall approximation, with all heavy (non-hydrogen) atoms of the Me(II)(OH)2 being part of a single rigid body (one rigid body per nanosheet). Thus, metal and oxygen atoms of the walls are able to move only in vertical direction z. Surface hydrogen atoms, being covalently bonded with oxygen atoms, had no constraints applied.”

9) Line 108-110: “The load was maintained by an external force -fz which was uniformly distributed over all non-hydrogen atoms of upper wall-piston.”

Dear authors, which type force was applied (for e.g is a umbrella potential)? What is the intensity of the applied force?

Response: Thank you for this comment. We applied the force as additional external constant force on non-hydrogen atoms of the upper wall-piston. We clarified this issue in the manuscript:

“The load was applied by an external force -fz, which was uniformly distributed over all non-hydrogen atoms of the upper wall-piston. The force was directed downward. Its magnitude was calculated as the product of the specified pressure and the area of the wall-piston surface. The external (applied) pressure p was varied in the range of 0.1-10 GPa.”

10) Line126-128:” Partial atomic charges of walls in the low-charged model system were 10 times lower compared to PACs of the first model. The PACs values used in the present study are listed in Supplementary Table S2.”

Dear authors, Could you justify the choice of reducing PACs by a factor of 10? Has this type of modification of the PACs already used in other works?

Dear authors could you provide other specification related to the computational protocol such as: The code used for the calculations? Ensemble (NVE NVT NPT)?

Response: Thank you for this comment. Reduced in this way, PACs provide electrostatic water-water interaction an order of magnitude more energetically favorable than water-surface interaction. We believed (and were right) that this was enough to "switch" from hydrophilic to hydrophobic surface.

The code and the models will be available on GitHub, the link is at the end of the manuscript (see the Data Availability Statement).

The computational protocol is now supplemented with following sentence in the revised manuscript:

“Temperature was constant and equal to 310 K in all simulations. The temperature was maintained by Nose-Hoover thermostat [64].”

Results

11) Line 172-193:” To investigate the structure and properties of nanoconfined water, as well as the influence of wall-water electrostatic interaction on water behavior under high pressure conditions, two moleculardynamic models of a slit-shaped nanopore filled with water were developed. Each model consists of an approximately 2 nm thick layer of water confined by two parallel metal hydroxide nanosheets located in the x-y plane (Figure 1). Along the x and y axes, periodic boundary conditions (PBC) were applied. In the first model, the class of naturally

occurred metal hydroxides Me(II)(OH)2 was chosen for the nanopore walls. The structural parameters of divalent iron hydroxide Fe(OH)2 with near medial lattice constant a [44] were chosen as a representative of layered hydroxides of divalent metals within the trigonal crystal system (Supplementary Table S1, and Supplementary Fig.S1). The second model of mineral is completely equivalent to the first one, except for modified partial atomic charges (PACs), whose values were reduced by a factor of ten compared to the first model (Supplementary Table S2). Henceforth we will refer to the first model as "Naturally charged host model" and the second model as "Low-charged host model". It should be noted that both models have the composition 36[2Me(II)(OH)2·7H2O], and, in total, each model is electrically neutral. In other words, two "host-guest" systems with the same atomic structure of the nanopore walls and the same number of "guest" water molecules were considered. The only difference between the two models is that one shows hydrophilic properties, and the other hydrophobic. A detailed description of the model and discussion of the advantages, disadvantages, assumptions, and limits of the chosen approach can be found in the Method and models section. ”

Dear authors, this part is too long and should be included in the methods part.

Response: Thank you for this comment. We moved this part to section “2. Materials and Methods”. We replaced this part in section “3. Results” with the following short introductory paragraph:

“In order to investigate the structure and properties of nanoconfined water, two molecular-dynamic models of water between parallel sheets of brucite-like metal hydroxide were utilized: (1) naturally charged host model; (2) low-charged (artificial) host model. The difference between the two models, as it can be seen further, is that the first one possesses hydrophilic properties, and the second one is hydrophobic. We analyzed nanoconfined water in the pressure range of 0.1-10 GPa.”

12) Line196-199:” This range corresponds, particularly, to tectonic plates at the characteristic depth interval from 3 km (upper crust) to 300 km (upper mantle), or to highly stressed contact spots in heavy-loaded tribounits.”

Dear authors could you add some references?

Response: Thank you for this comment. We added two references:

“This range particularly corresponds to tectonic plates at the characteristic depth interval from 3 km (upper crust) to 300 km (upper mantle) [20], or to highly stressed contact spots in heavily loaded tribological couples [67].”

[20] Hwang, H.; Seoung, D.; Lee, Y.; Liu, Z.; Liermann, H.-P.; Cynn, H.; Kao, C.-C.; Mao, H.-K. A role for subducted super-hydrated kaolinite in Earth’s deep water cycle. Nat. Geosci. 2017, 10, 947–953. https://doi.org/10.1038/s41561-017-0008-1.

[67] Voevodin, A.A.; O'neill, J.P.; Zabinski, J.S. Nanocomposite tribological coatings for aerospace applications. Surf. Coat. Technol. 1999, 116, 36–45. https://doi.org/10.1016/S0257-8972(99)00228-5.

13) Line 199:” The temperature was constant and equal to 310 K in all simulations.”

Dear authors, this part should be included in the methods part.

Response: Thank you for this comment. This part was moved to section “2. Materials and Methods”.

14) Line200-206:” A comprehensive analysis of water structure and properties was undertaken during the simulation runs: estimates of water density, uniaxial compressibility (Figure 2a, and further Figure 4), spatial arrangement of water molecules (Figures 2b, Figure 3 and Supplementary Figure S3) including the radial distribution functions (RDF) (Supplementary Figures S4 and S5). Further, the number of hydrogen bonds and the self-diffusion coefficient (Figure 5a,c and Supplementary Figures S6-S7) were numerically obtained as functions of applied pressure and nanopore surface polarity.

Dear authors, this part should be included in the methods part. Maybe you could create a new sub-section related to the analysis.

I suggest replacing the images with high definition images

Response: Thank you for this comment. We have moved this part to the end of section “2. Materials and Methods".

High-resolution images are submitted separately from the manuscript. Quality of the images in the manuscript is reduced by Word.
